

# Microbial response to deliquescence of nitrate-rich soils in the hyperarid Atacama Desert

Felix L. Arens[1*], Alessandro Airo[1,2], Christof Sager[1,2], Hans-Peter Grossart[3,4], Kai Mangelsdorf[5], Rainer U. Meckenstock[6], Mark Pannekens[6], Philippe Schmitt-Kopplin[7,8], Jenny Uhl[7], Bernardita Valenzuela[9], Pedro Zamorano[10], Luca Zoccarato[3,11,12], Dirk Schulze-Makuch[1,3,13]

[1]Technische Universität Berlin, Zentrum für Astronomie und Astrophysik, 10623 Berlin, Germany
[2]Museum für Naturkunde, Leibniz-Institut für Evolutions- und Biodiversitätsforschung, 10115 Berlin, Germany
[3]Department of Experimental Limnology, Leibniz-Institute of Freshwater Ecology and Inland Fisheries, 16775 Stechlin, Germany
[4]Institute for Biochemistry and Biology, Potsdam University, 14469 Potsdam, Germany
[5]Section Organic Geochenistry, Helmholtz Centre Potsdam GFZ German Research Centre for Geosciences, Potsdam, Germany
[6]Environmental Microbiology and Biotechnology, University of Duisburg-Essen, 45141 Essen
[7]Helmholtz Zentrum München, Research Unit Analytical Biogeochemistry, 85764 Neuherberg, Germany
[8]Technische Universität München, Chair of Analytical Food Chemistry, 85354 Freising, Germany
[9]Laboratorio de Microorganismos Extremófilos, Instituto Antofagasta, Universidad de Antofagasta, Antofagasta 1240000, Chile
[10]Departamento Biomédico, Facultad de Ciencias de la Salud, Universidad de Antofagasta; Antofagasta 1240000, Chile
[11]Core Facility Bioinformatics, University of Natural Resources and Life Sciences (BOKU), 1190 Vienna, Austria
[12]Institute of Computational Biology, University of Natural Resources and Life Sciences, 1180 Vienna, Austria
[13]GFZ German Research Centre for Geosciences, Section Geomicrobiology, 14473 Potsdam, Germany

* *Correspondence to*: Felix L. Arens (f.arens@tu-berlin.de)

## ABSTRACT

Life in hyperarid regions has adapted to extreme water scarcity by using salt deliquescence. Here, we investigated newly discovered deliquescent soil surfaces in the Atacama Desert, containing substantial amounts of nitrates, to evaluate their habitability for microorganisms. We characterized the environment regarding water availability and biogeochemistry. Microbial abundances and composition were determined by cell cultivation experiments and 16S rRNA gene sequencing while microbial activity was assessed by analyzing ATP, PLFA, and the molecular composition of organic matter. Our findings reveal that while the studied hygroscopic salts provide temporary water, microbial abundances and activities are lower than in non-deliquescent soil surfaces. Intriguingly, the deliquescent crusts are enriched in geochemically degraded organic matter. We conclude that high nitrate concentrations in the hyperarid soils suppress microbial activity but preserve eolian-derived biomolecules. These insights are important for assessing the habitability and searching for life in hyperarid environments on Earth and beyond.



## 1 INTRODUCTION

The Atacama Desert is one of the driest and oldest deserts on Earth with hyperarid conditions established in the Oligocene (Dunai et al., 2005; Jordan et al., 2014). Over the last two decades, the Atacama Desert has been intensively studied as a Mars analog and for the dry limits of life along aridity gradients progressing towards hyperaridity (Quade et al., 2007; Schulze-Makuch et al., 2018). Vegetation density decreases with increasing aridity until vascular plants become absent in the absolute desert (Quade et al., 2007). It has long remained unclear whether there is active life or whether recovered DNA is only blown in from the atmosphere and slowly decaying (Navarro-Gonzalez et al., 2003; Lester et al., 2007). However, later studies showed that microbial life can indeed survive and temporally thrive within the hyperarid core of the Atacama Desert (Warren-Rhodes et al., 2006; Wierzchos et al., 2006; Connon et al., 2007; Wierzchos et al., 2012; Schulze-Makuch et al., 2018; Hwang et al., 2021; Schulze-Makuch et al., 2021).

With increasing aridity, life retreats from the surface into the subsurface. Photosynthesis-based microbial communities inhabit hypolithic and endolithic habitats under translucent rocks and crusts or within their pore space (Warren-Rhodes et al., 2006; Wierzchos et al., 2011). These micro-environments provide shelter against UV-radiation while receiving sunlight and buffering evaporation and temperature fluctuation. These ecosystems can be found widely in the arid part of the Atacama Desert and even sporadically in the hyperarid region (Warren-Rhodes et al., 2006). The one of the last islands of habitability towards the dry limit of life are found inside surficial salt crusts (Wierzchos et al., 2006; Davila and Schulze-Makuch, 2016; Schulze-Makuch et al., 2021). These can provide liquid water through deliquescence of salt, i.e., halite, absorbing water vapor from humid air (>75 % relative humidity (RH)) and forming a saturated brine on the salt crust surface and within the soil pore space (Davila et al., 2013; Robinson et al., 2015; Maus et al., 2020). In contrast to rain and fog, deliquescence might be the last source of liquid water, enabling microbial colonization in a unique ecological sequence towards increasing aridity (Davila and Schulze-Makuch, 2016).

In the Atacama Desert, salt crusts are commonly found in dried-out saline lakes, locally called salars, with prominent salt aggregates at the surface (Stoertz and Ericksen, 1974). The so-called salt nodules are formed by cycles of deliquescence and efflorescence and are superimposed by eolian erosion (Artieda et al., 2015). They are mainly composed of halite with varying fractions of gypsum and lithic detrital clasts (Wierzchos et al., 2006; Robinson et al., 2015; Schulze-Makuch et al., 2021). Apart from salars, salt accumulations are generally found within the Atacama Desert in the subsurface of alluvial deposits, which have accumulated over millions of years (Ericksen, 1981; Ewing et al., 2006). The prolonged hyperarid conditions resulted in atmospheric salt accumulation and a post-depositional separation within the soil column through rare rain water infiltration (Ewing et al., 2006; 2008; Arens et al., 2021). As a result, highly soluble and hygroscopic NaCl and NaNO$_3$ migrate deeper into subsurface horizons, locally called *caliche*. The soil above is dominated by sulfate. Close to the surface, the soil is exceptionally porous (*chusca*) and becomes more firmly cemented in the subsurface (*costra*) (Ericksen, 1981). Thermal stress and salt dehydration lead to cracks which can develop into sand wedges that shape the typical hexagonal and orthogonal soil polygons in the Atacama Desert (Ewing et al., 2006; Pfeiffer et al., 2021; Sager et al., 2021) (Fig. 1).



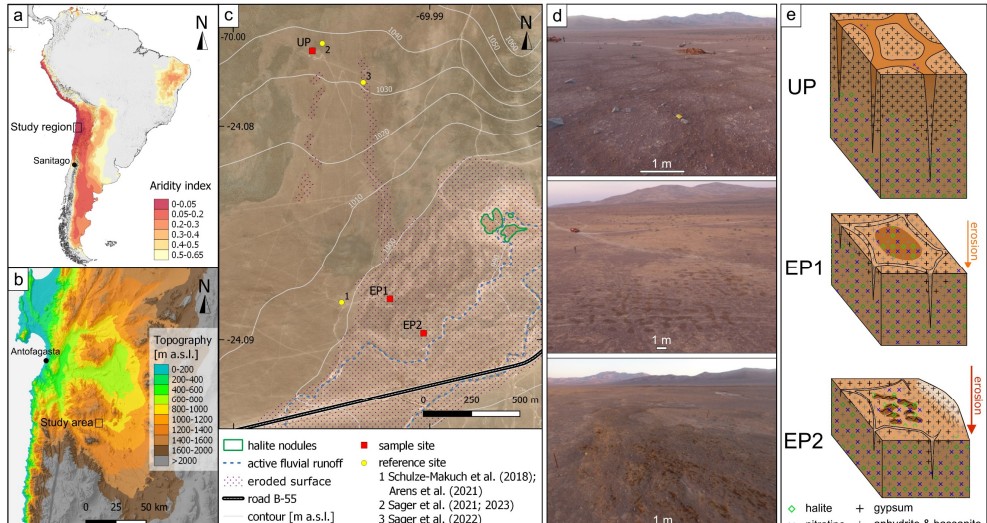

Figure 1: Overview of the study area. a) Map of South America with color code for the aridity index with <0.05 being hyperarid (Zomer et al., 2022). b) Topographic map of the study region, with the Yungay valley, 60 km southeast of Antofagasta, where the study area is located. c) Landsat-8 satellite image of the study area with 10 m interval isohyets, showing the three sample sites and relevant reference sites. The purple dotted area marks surface erosion and the blue dashed line indicates main run-off channels, active during the last major rain events (2017). The nearest observed salt nodules are outlined in green. d) Aerial photos of the study sites during morning hours. e) Sketches of the soil structures at each site with salt distribution. Darker surface areas indicate potential deliquescence.

Further, local eolian erosion can lead to the exposure of salt-rich subsurface down to the *caliche* horizon (Sager et al., 2022). Analogous to halite nodules, salt-encrusted surfaces can form here, composed of sulfate, chloride, and nitrate salts, that develop similar efflorescent morphologies (Fig. 2). While halite-rich soil crusts have been shown to be inhabited by microbes (Wierzchos et al., 2006), the potential role of nitrate-rich soil crusts as microbial habitats remains unclear. This study aims to characterize hygroscopic nitrate-rich soil crusts within the hyperarid Atacama Desert, employing an interdisciplinary approach that integrates geochemical, biogeochemical, and microbiological methods. The goal is to unravel the significance of nitrates for microbial life in one of the most arid regions on Earth, serving as an outstanding Martian analog. These hypersaline environments are especially interesting for the search for life on Mars where nitrates have been detected (Stern et al., 2015), as these may provide a last refuge for putative Martian organisms (Davila and Schulze-Makuch, 2016) and could serve as excellent candidates for the preservation of biosignatures (Fernández-Remolar et al., 2013).

## 2 METHODS

### 2.1 Study area and sampling

The here investigated soil surfaces are located in the Yungay valley within the hyperarid Atacama Desert, Chile (Fig. 1a, b) (UP: 24.076S 69.995W; EP1: 24.088S 69.992W; EP2: 24.090S 69.991W). The sample sites are located on a distal part of an alluvial fan, which developed polygonal patterned grounds on its surface (Fig. 1c). Deliquescence-induced water uptake capacities and potential changes in microbial activity were evaluated by taking samples in the morning (potentially moist) and in the evening (dry). At each sampling site, surface samples in 0–5 cm depth were taken in the deliquescence affected area and in adjacent areas which were not affected by deliquescence. Roughly 100 g sample material for geochemical analysis were collected in PE bags. Triplicate samples for water activity and content were stored in 100 mL glass bottles with PTFE sealed lids at 4 °C until analysis. Biological samples were sampled in triplicates in 50 mL centrifuge tubes and stored at −20 °C until analysis. Precautions were taken to keep all samples sterile and to avoid cross-contamination by wearing nitrile gloves as well as by wiping and flaming the sampling tools using ethanol before each use.



## 2.2 Environmental monitoring

Temperature and RH of the air (1 m above ground) in the study area was recorded between 2018 and 2019 using environmental loggers U23-001 by Onset (USA). Soil electrical conductivity was measured on selected surfaces in 0-5 cm depths using a CR10 (Campbell Scientific, USA). Aerial images were taken by a DJI Phantom 4 unmanned aerial vehicle and later processed into orthophotos and DEMs with Agisoft Metashape Pro software. Field images were calibrated with SpyderCHECKR®24 (datacolor, Switzerland) and post-processed for color correction with checkr24 (datacolor, Switzerland) software.

## 2.3 Water activity and content analysis

Triplicate samples were analyzed for water activity and content analysis. The water content of the collected samples was determined by the weight loss after drying at 60 °C for 24 h to avoid the dehydration of gypsum. The water activity was analyzed with a LabMaster-aw neo (Switzerland) equipped with an electrolytic sensor.

## 2.4 Geochemical and mineral analyses

### 2.4.1 Mineral analysis

The bulk mineralogy was analyzed via powder XRD. 5 g sample aliquots were dried at 60 °C and ground to powder. XRD analysis was performed by using a D2 Phaser (Bruker, USA) powder diffractometer. The X-ray source is a Cu Kα radiation (K-alpha1= 1.540598 Å, K-alpha2=1.54439 Å) with a performance of 30 kV and 10 mA. A step interval of 0.013° 2Θ with a step-counting time of 20 s was used in a scanning range from 5° to 90° 2Θ. Evaluation was conducted semi-quantitatively using the "Powder Diffraction File Minerals 2019" (International Centre of Diffraction Data) together with the software High Score from PANalytical (Netherlands).

### 2.4.2 Ion chromatography

Anionic species ($Cl^-$, $NO_3^-$, $SO_4^{2-}$) were measured by ion chromatography (DIONEX DX-120 ion chromatograph, Thermo Fisher Scientific Inc., USA). Samples were dried at 60 °C, sieved dry to <2 mm grain size, and leached in duplicates with a 1:10 ratio (sample:water (w/w)). Samples were measured in duplicates and blanks were measured alongside the samples for quality control.

### 2.4.3 Elemental analysis

Total carbon, nitrogen, and sulfur were measured on homogenized, powdered samples with a Vario Max CNS (Elementar GmbH, Germany) at 1140 °C combustion temperature. TOC was measured on a Vario Max C by combustion at 600 °C. Measurements were performed in duplicates with 1 g of sample and standards were used to determine detection limits of 0.01 wt% for C, N, S and 0.03 wt% for TOC. TIC was calculated as the difference between total carbon and organic carbon.

## 2.5 Biological analyses

### 2.5.1 Adenosine triphosphate (ATP) analysis

Sediment samples were placed in a sterile autoclave bag and crushed into smaller pieces (up to a maximum diameter of approximately 1 cm) using a hammer. 6 g of sediment or crushed rock samples were introduced into a 50 mL centrifuge tube, and 5 mL of ice-cold sodium phosphate buffer (0.12 M $Na_2HPO_4$, $NaH_2PO_4$, pH = 8.0) was added. Samples were shaken on an orbital shaker for 5 min at 150 rpm, cooled on ice for 3 min, and shaken again for another 5 min. Samples were then centrifuged at 4 °C and 500 $g$ for 10 min. The supernatants, which contain the tATP, were recovered in a 15 mL centrifuge tube, and 1 mL of sodium phosphate buffer was added to the sediment samples. The procedure was repeated 3 times and supernatants were collected. This was done separately for the tATP and iATP. For the iATP, the collected suspensions were centrifuged at 4 °C and 4,600 $g$ for 60 min. Cell pellets containing iATP were re-suspended in 4 mL of sodium phosphate buffer and the particles in the solution were allowed to settle for approximately 30 min before samples were subjected to ATP analysis. All samples were processed in triplicates. ATP was quantified using the luciferase-based BacTiter-Glo™ Microbial Cell Viability Assay (Promega, USA). Measurements for the iATP were carried out according to the



manufacturer's protocol, using a 6-point calibration curve with ATP concentrations ranging from 10 pM to 1μM
in a 0.12 M sodium phosphate buffer. For the tATP a 5-step standard addition with 1, 2, 3, 4 μL of 0.1 μM ATP
was applied to avoid matrix effects potentially caused by the dissolved soil salts (supplementary information S6).
Finally, 100 μL of sample solution, blank, or standard were mixed with 100 μL of BacTiter-Glo$^{TM}$ reagent, which
was prepared on the day before measurement and kept at room temperature until measurements were performed.
5 minutes after mixing, luminescence was recorded using a Glomax 20/20 luminometer (Promega, USA).

### 160 *2.5.2 Phospholipid fatty acid (PLFA)*

PLFA extraction and subsequent analysis were conducted with the procedure described in detail by (Zink and
Mangelsdorf, 2004) and (Sager et al., 2023). PLFAs were obtained from intact membrane phospholipids by
applying an ester cleavage procedure (Müller et al., 1990). Hereby, the phospholipid linked fatty esters are directly
transformed into their respective fatty acid methyl esters (PLFAs) using trimethylsulfonium hydroxide.
Subsequently, the PLFAs were measured on a trace gas chromatograph (GC) 1310 (Thermo Scientific, USA)
coupled to a TSQ 9000 mass spectrometer (MS) (Thermo Scientific, USA). The GC was equipped with a cold
injection system operating in the splitless mode and a SGE BPX 5 fused-silica capillary column (50 m length,
0.22 mm ID, 0.25 μm film thickness) with initial temperature of 50 °C (1 min isothermal), heating rate 3 °C min$^{-1}$
to 310 °C, held isothermally for 30 min. Helium was used as carrier gas with a constant flow of 1 mL min$^{-1}$. The
injector temperature was programmed from 50 to 300 °C at a rate of 10 °C s$^{-1}$. The MS operated in electron impact
mode at 70 eV. Full-scan mass spectra were recorded from m/z 50 to 650 at a scan rate of 1.5 scans s$^{-1}$. A blank
was prepared and measured alongside the samples for quality control.

### 173 *2.5.3 16S rRNA gene sequencing*

DNA extraction of soil samples was performed based on a slightly modified protocol of Nercessian et al.,
(Nercessian et al., 2005) with sample aliquots of 5 g. In brief, cell lysis was performed using glass beads (100–500
μm) in the presence of lysozyme, proteinase K and cetyltrimethyl ammonium bromide (CTAB). DNA purification
was facilitated by the addition of Phenol-Chloroform and polyethylene glycol (PEG) (Neubauer et al., 2021). The
V3-V4 region of the 16S rRNA was amplified using the S-D-Bact-0341-b-S-17 / S-D-Bact-0785-a-A-21 primer
pair (Mitra et al., 2013), while library preparation and sequencing were carried out on an Illumina MiSeq
instrument (Illumina, USA).
Demultiplexing, removal of primer and adapter sequences were performed using Cutadapt v3.7 (Martin, 2011).
Fastq files are deposited in the SRA. Additional quality filtering and trimming, identification of unique amplicon
sequence variants (ASVs) and paired reads merging were performed using the DADA2 v1.20 (Callahan et al.,
2016) following the standard pipeline with default values (we set pool = T for the dada() function and
method = "consensus" for the removeBimeraDenovo() function). Taxonomy was assigned to ASVs using SINA
v1.7.2 (Pruesse et al., 2012) against the SILVA reference database (SSU NR 99 v138.1; (Quast et al., 2012)).
ASVs having less than five total reads or which occurred in less than three samples were removed from
downstream analyses. Alpha and beta diversity analyses were performed in R phyloseq package (McMurdie and
Holmes, 2013). Alpha diversity (Chao1) was calculated and the function *estimateR* (R package vegan) was used
to estimate ASV richness as it accounts for differences in library sizes. For the Principal Coordinate Analysis
(PCoA), ASV counts have been centered-log-ratio transformed using the function *decostand* (method = "rclr",
package vegan). The Aitchison distance was then obtained with the vegan function vegdist (method = "euclidean",
R package vegan) and the PCoA was plotted using *plot_ordination* (method = "PCoA", R package phyloseq,
(Wickham et al., 2016)). Distance-based linear modeling was performed using normalized environmental variables
(function decostand, method = "normalize"), and significant variables were visualized via canonical analysis of
principal coordinates (CAP) plot. The CAP was carried out to relate bacterial communities to different
environmental variables (including EC, gypsum, Cl$^{-}$, NO$_3^{-}$, ATP, TOC).

### 198 *2.5.4 Cell cultivation experiments*

Microbial cell abundance was estimated by carrying out cultivation experiments following the protocol by (Knief
et al., 2020). In triplicates, 5 g sample aliquot was suspended in 25 mL of sterile phosphate buffer solution
(120 mM, pH = 8) and incubated for 30 min at 60 rpm at room temperature in a shaker (LabNet, USA) followed
by 2 min ultrasonication in a water bath (Emlasonic S 30H, Germany). 100 μL of the obtained suspensions were
spread in triplicates on agar plates. Nutrient broth medium was used for the growth of bacterial cells consisting of
3 g L$^{-1}$ yeast extract, 3 g L$^{-1}$ peptone, and 15 g L$^{-1}$ agar. Plates were incubated at room temperature and evaluated



for bacterial growth after 4 weeks by counting the colony forming units (CFUs). Bacterial genomic DNA of
individual CFUs was extracted using the Wizard Genomic DNA Purification Kit (Promega, Madison, WI, USA)
and amplified through PCR targeting the universal 16S rDNA region with bacterial primers 27F and 1525R
(Altschul et al., 1997). PCR reactions utilized the Go Taq Green Master Mix kit (Promega, Valencia, CA, USA),
with cycling conditions including an initial denaturation at 95 ºC for 5 min, followed by 35 cycles of denaturation
(95 ºC for 30 s), annealing 55 ºC for 30 s, and extension 72 ºC for 1.5. PCR products' integrity was confirmed
through gel electrophoresis, and the amplicons were sequenced at Macrogen (Republic of Korea) and analyzed for
comparison with GenBank (NCBI) sequences.

### 2.5.5 Profiling organic matter via FT-ICR-MS

The same extraction and analytical protocol as for similar studies in the region were used to gain comparability
(Schulze-Makuch et al., 2018; Schulze-Makuch et al., 2021). Mass spectra were acquired in negative electrospray
ionization (ESI) mode using a SolariX Qe FT-ICR-MS equipped with a 12 T superconducting magnet and coupled
to an Apollo II ESI-source (Bruker Daltonics, Germany). Methanolic soil extracts were continuously infused with
a flow rate of 120 μL h$^{-1}$. Spectra accumulated 500 scans within a mass range of 147 to 1000 m/z. An internal
calibration was performed with a mass accuracy of <0.1 ppm, and peaks with a signal to noise ratio >6 were picked.
Formula assignment was performed with in-house written software (NetCalc) using a network approach to
calculate chemical compositions containing carbon, hydrogen, and oxygen, as well as nitrogen and/or sulfur. The
mass accuracy window for the formula assignment was set to ±0.5 ppm, and the assigned formulas were validated
by setting sensible chemical constraints (N rule; O/C ratio ≥1; H/C ratio ≤ 2n + 2 (maximum possible carbon
saturation, with n defined as CnHn+2 for any formula), double bond equivalents) in conjunction with isotope
pattern comparison. Results were visualized using van Krevelen diagrams in which the hydrogen to carbon ratio
(H/C) was plotted against the oxygen to carbon ratio (O/C). The different bubble sizes represent the intensity of
the characteristic molecular formula within the respective sample.

## 3 RESULTS

The influence of deliquescence on soil habitability was investigated on three selected sampling sites on polygonal
soils: uneroded (UP), moderately eroded (EP1) and strongly eroded (EP2), where repeated deliquescence was
observed in varying intensities (Fig. 1, 2). This was most pronounced at the EP2 site, which we chose as our
primary target.

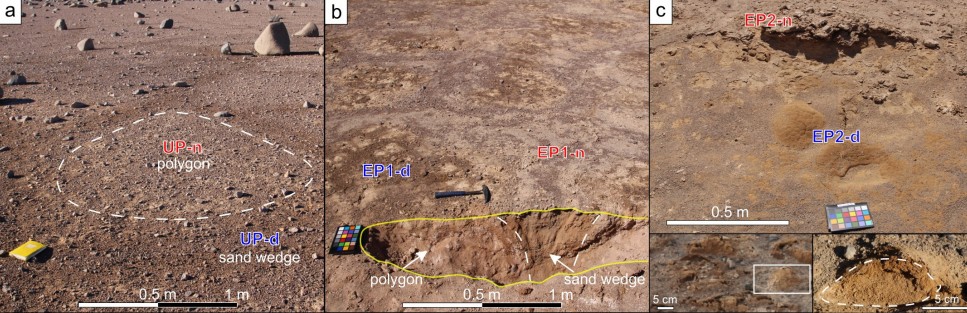


Figure 2: Images of the sample sites. Bright soil colors indicate sulfates, dark soil colors indicate nitrates and chlorides. a) UP
site with the darker sand wedge surface, enclosing the brighter polygon surface. Example polygon outline with white dashes.
b) EP1 site with dark polygon surface, surrounded by bright sand wedges. Excavation pit outlined yellow, border between
polygon and sand wedge marked with white dashes. c) EP2 site with small troughs formed by eolian erosion exposing nitrate-
and chloride-rich soil which appear dark brown. Remains of the overlying *chusca* are visible in the background. Left inlet:
detailed image of efflorescent morphologies within EP2-d. White box indicates area of right inlet: cross section (white dashed
line) of an efflorescence dome. Moisture reached a few centimeters into the ground. The soil below remained dry.
Soil moistening by deliquescence was observed in the morning on the surface of the polygons (EP1-d and EP2-d;
"d" for deliquescent"), as well as on isolated patches of sand wedge surfaces within uneroded polygonal soils (UP-
d), surrounded by otherwise dry surfaces (UP-n, EP1-n, EP2-n; "n" for non-deliquescent") (Fig. 3).



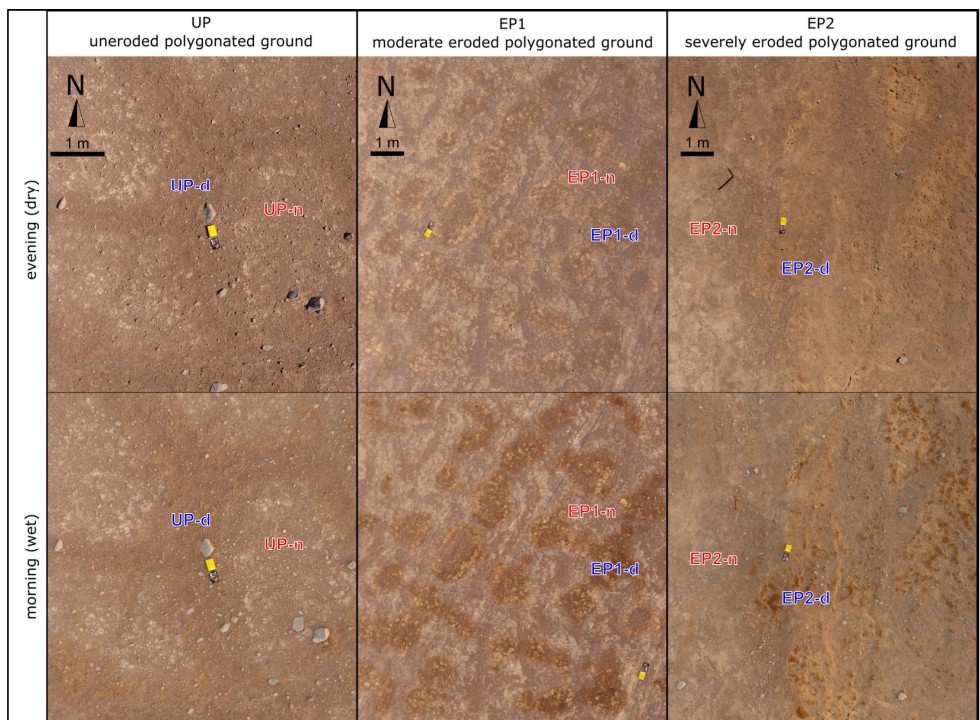

Figure 3: Aerial photos of the study sites during the evening and the morning that were corrected by color calibration chart. At UP darker areas in the morning occurred sporadically on the surface of the sand wedges. At the eroded polygon sites the surface of the polygons is darker in the morning, at EP1 uniform and at EP2 especially the elevated domes and crusts (Fig. 2c).

The ambient conditions in the study area are strongly determined by the diurnal cycle (Fig. 4a). During the night, RH reached 90 % and air temperature dropped to 5 °C, while during the day, RH decreased to 10 % with air temperature increasing to 40 °C. The *in situ* soil electrical conductivity ($EC_{in\ situ}$) is a function of salinity and moisture and measurements over time can indicate moistening and desiccation of the soil. At EP2-d during 14th and 15th of March 2019, $EC_{in\ situ}$ gradually increased during the night, indicating brine formation, and decreased rapidly after sunrise, indicating soil desiccation. In contrast, the sensors at EP2-n continuously detected low $EC_{in\ situ}$ (Fig. 4b), but also measured a minor increase during the morning, which can indicate the formation of morning dew. Moisture was observed down to ~5 cm depth, and below the soil remained dry. The water activity ($a_w$) remained generally low with $a_w$ <0.5 except for the EP2-d in the morning (7:30 local time), with $a_w$ = 0.71 (Fig. 4c). Here, water content was most elevated, highlighting the high deliquescence potential of this site. In the EP1-d, the water uptake during the night was not as prominent. Moreover, the water content in the evening sample (19:30 local time) remained elevated. This suggests the presence of hydrated minerals like mirabilite which can dehydrate during the drying process at 60 °C. However, these were not found with X-ray diffraction (XRD). At the UP-d site, no significant water uptake could be detected with the applied method (Fig. 4d).

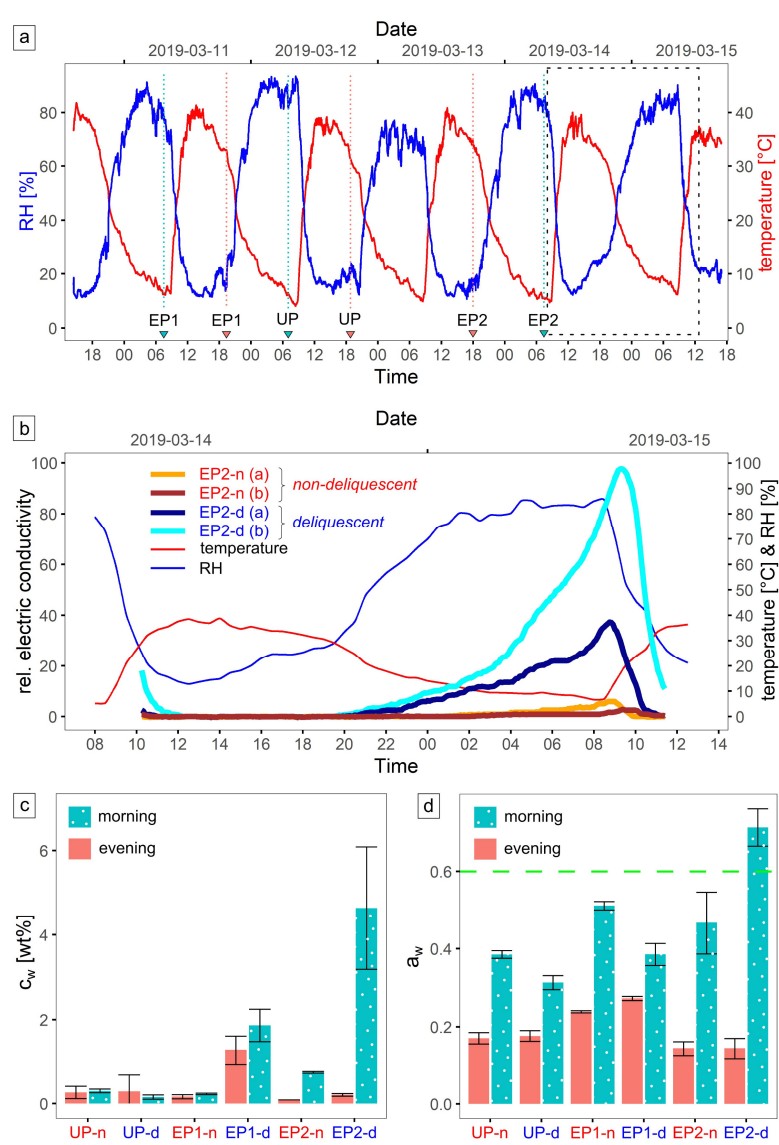

Figure 4: Environmental monitoring data. a) Air temperature and relative humidity (RH) in the study area recorded during the sampling campaign with the sampling time (local time UTC−3 h), marked by blue triangles (morning) and red triangles (evening), and the zoom-in area (dashed box) for b). b) Relative electric conductivity (EC$_{in\ situ}$) of the surface (0-5 cm depth) at EP2 site for a day cycle. The deviation between the replicate measurement can be manyfold, either by different salt composition or texture of the soil in the measurement volume or by poor electrode contact. c) Water content (c$_w$) and d) water activity (a$_w$) for each sample site in the evening (18:00) and in the morning (7:30). The green dashed line is the limit for microbial activity (Stevenson et al., 2015). Uncertainties derived from triplicate samples.

For the geochemical analysis the samples taken during the morning were selected. The XRD and ion chromatography (IC) analysis revealed that samples, which experienced intense deliquescence (EP1-d and EP2-d), contain up to 50 g kg$^{-1}$ chlorides in the form of halite (NaCl) and up to 110 g kg$^{-1}$ nitrates in the form of nitratine (NaNO$_3$) (Fig. 5a, b). In the samples from UP-d with minor and isolated deliquescence spots, XRD did not detect any salts, but the more sensitive IC detected low concentrations of nitrate (8 g kg$^{-1}$) and chloride (3 g kg$^{-1}$). The non-deliquescence sites (UP-n, EP1-n, EP2-n) are dominated by sulfates, mainly gypsum




(CaSO$_4$×2H$_2$O) and minor amounts of anhydrite (CaSO$_4$) or bassanite (CaSO$_4$×0.5H$_2$O). In the deliquescent soils,
gypsum, anhydrite, and bassanite have also been detected, but in lower quantities. The quantity of sulfates is better
represented in the semi-quantitative XRD data; as for the IC analysis, samples were leached with a 1:10 (soil to
water) ratio, being unable to dissolve entirely calcium sulfate (water solubility ~2 g L$^{-1}$) (Fig. 5a, b). The sand
wedges at UP are salt poor, however, they contain small amounts of chloride and nitrate up to 10 g kg$^{-1}$. Besides
the salts, EP2 samples, especially EP2-d, contained detectable amounts of phyllosilicates and calcite (Fig. 5a).
Elemental analysis of nitrogen (N) and sulfur (S) for the EP2 samples supports the XRD results, showing nitrogen
enrichment in the EP2-d samples and levels close to the detection limit (0.1 g kg$^{-1}$) in the EP2-n samples. In
contrast, these samples are more concentrated in sulfur while the deliquescent samples (EP2-d) have comparably
low levels (Fig. S2). Carbon (C) is found in the soil as organic matter and as carbonate, given as total organic
carbon (TOC) and total inorganic carbon (TIC), respectively (Fig. 5c). TIC is most concentrated in the EP2
samples, with up to 5.8 g kg$^{-1}$, while TOC can be detected where deliquescence was observed predominantly (EP2-
d & EP1-d), reaching values of up to 3.7 g kg$^{-1}$. In the surrounding soils, organic carbon was below the detection
limit (0.1 g kg$^{-1}$).

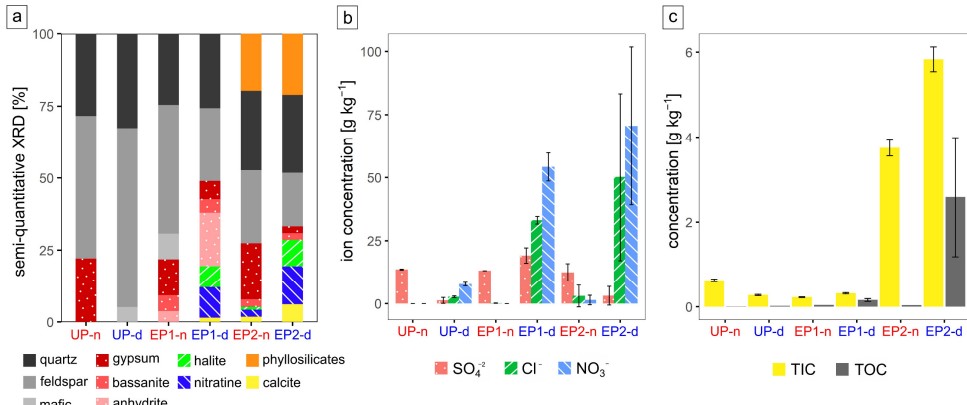

Figure 5: Geochemical data. A) semi-quantitative mineralogical composition by XRD of bulk samples. B) Concentration of
the main water-soluble anions. C) Total carbon concentration shown as total organic carbon (TOC) and total inorganic carbon
(TIC). Uncertainties derived from triplicate samples.
The collected biological data is overall very sparse, reflecting the harsh conditions in this extreme environment.
Adenosine triphosphate (ATP) is the ubiquitously used energy source by life and can be utilized as an indicator of
microbial activity (Blagodatskaya and Kuzyakov, 2013). The total ATP concentrations (tATP) in our samples were
extremely low, with values of 1 pmol g$^{-1}$ sediment or even lower, reflecting the extreme conditions for life in the
Atacama Desert (Fig. 6a). The intracellular ATP (iATP), extracted from intact cells, is only a small fraction of the
tATP and is overall lower in the deliquescent soils compared to the surrounding non-deliquescent soils (Fig. 6b).
Significant turnover rates during morning and evening are not visible.
The following biological analyses were employed on the samples which were sampled in the morning.
Phospholipid fatty acids (PLFA) are indicative for soil habitability and cell viability as they are the main
components of bacterial membranes that can easily degrade after cell death (Connon et al., 2007). Additionally,
they can be used to analyze the general microbial community on a broad taxonomic level (Mangelsdorf et al.,
2020). For comparison between the deliquescent and non-deliquescent surfaces, two replicate samples from EP2-
d (EP2-d a, EP2-d b) and from EP2-n and UP-n one sample each were selected for PLFA analysis. PLFAs were
found in all investigated samples with concentrations above the blank (37 pmol g$^{-1}$). The deliquescent soils with
nitrate and chloride salts contained less PLFAs (160–308 pmol g$^{-1}$) than the non-deliquescent sulfate-cemented
soils (430–581 pmol g$^{-1}$) (Fig. 6c). This trend has also been found in the PLFA diversity, where the deliquescent
samples have 6 and 10 different PLFAs compared to 15 and 20 in the non-deliquescent samples (Fig. 6d). In the
overall inventory, the normal saturated (58 %) and the monoenoic fatty acids (24 %) were most abundant and were
found together with the polyenoic fatty acids (3 %) in all samples. The terminally branched saturated acids were
found in the low saline, non-deliquescent UP-n and EP2-n samples and the dicarboxylic fatty acids, known for
*Acidobacteria* membrane, are exclusively detected in the high saline, deliquescent EP2-d samples.




The cultivation experiments conducted with the EP2 samples yielded colony forming unit (CFU) counts in the
order of $10^2$–$10^3$ cells g$^{-1}$ soil (Fig. 6e). The CFU values of EP2-d are on average lower compared to the EP2-n
samples, indicating lower bacterial abundance in the deliquescent soils. Additionally, 16S rRNA gene sequencing
was performed on individual colonies identifying eight different genera in the surface soil, five in EP2-d samples
and four in EP2-n samples. Bacteria of the genus *Advevella*, *Microbacterium*, *Pseudomonas* and *Rhizobium* were
found exclusively in the EP2-d, and the genus *Bacillus, Hydrogenophaga* and *Variovorax* exclusively in EP2-n
(Fig. 6f, Table S1).

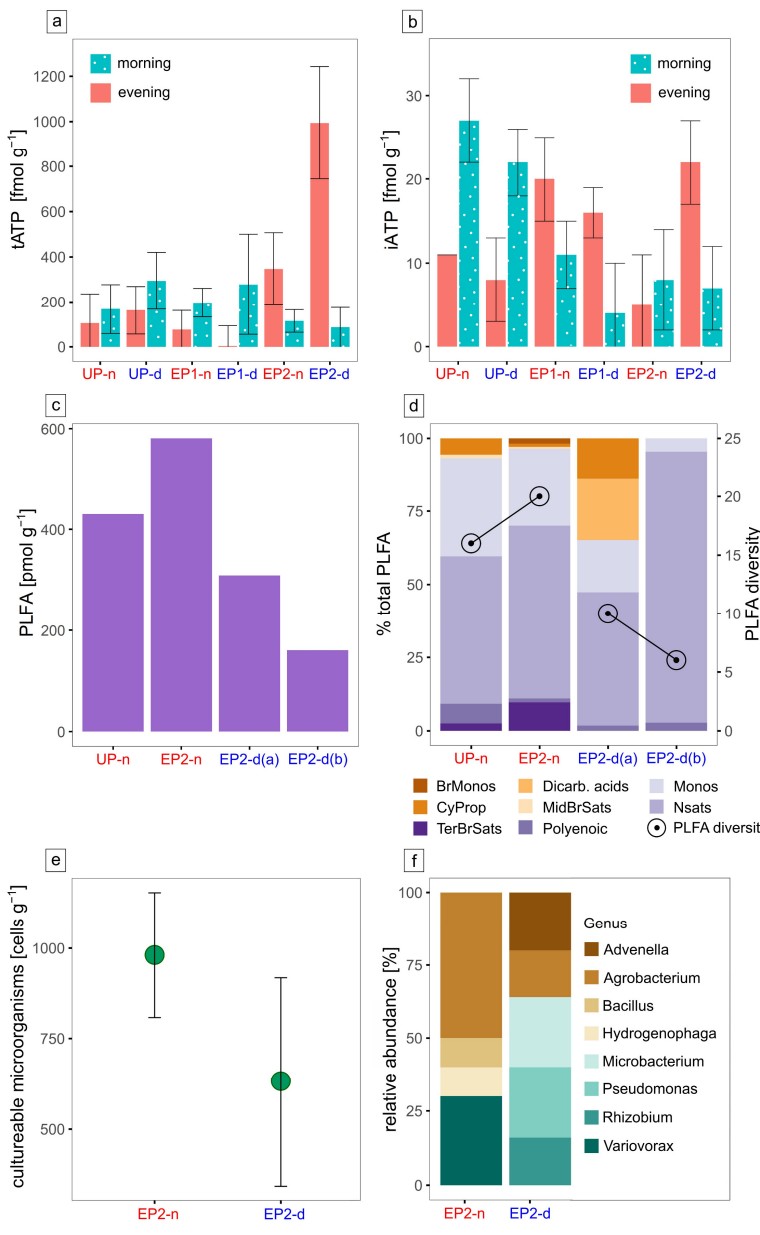


Figure 6: Microbial activity data. ATP concentration at all sampling sites during morning and evening hours split in a) total
ATP (tATP) and b) intracellular ATP (iATP) concentration. c) PLFA concentration. d) The relative abundance of different
PLFA groups including branched monoenoic (BrMonos), dicarboxylic acids (dicarb. acids), monoenoic (Monos), cyclopropyl





(CyProp), mid-chain branched saturated (MidBrSats), terminally branched saturated (TerBrSats), normal saturated (Nsats) and polyenoic fatty acids, as well as the PLFA diversity (number of different PLFA). e) Dot plot of the cultivation experiment data with colony forming units (CFUs) per sample weight, uncertainties derive from sample triplicates. f) 16S rRNA sequences of the cultivated microbes on genus level. PLFA analysis and culturing experiments were focused on the EP2 site where most intense deliquescence occurred.

Culture-independent 16S rRNA gene PCR amplicon sequencing using the bulk soil sample was challenging due to very low DNA concentration resulting from low microbial abundance, which prevented a statistically significant distinction between deliquescent and non-deliquescent soils. The alpha diversity is slightly higher for the deliquescence samples which supports the cultivation experiment results (Fig. S3), but the canonical analysis of principal coordinates is inconclusive (Fig. S4).

To gain a more comprehensive understanding of the increased organic matter in the deliquescent samples and to compare it with the non-deliquescent samples, organic molecules were measured via direct injection electrospray ionization Fourier transform ion cyclotron resonance mass spectrometry (ESI(−) FT-ICR-MS). Each mass signal was assigned to its corresponding molecular composition and classified as CHO, CHOS, or CHNO species. The comparison of uneroded and eroded soils differs in terms of the number of annotated elemental compositions. The results show a higher abundance of CHO and CHNO molecular features is found in the intense deliquescence soil surface of the eroded polygon sites (EP1-d, EP2-d) (Fig. 7a).

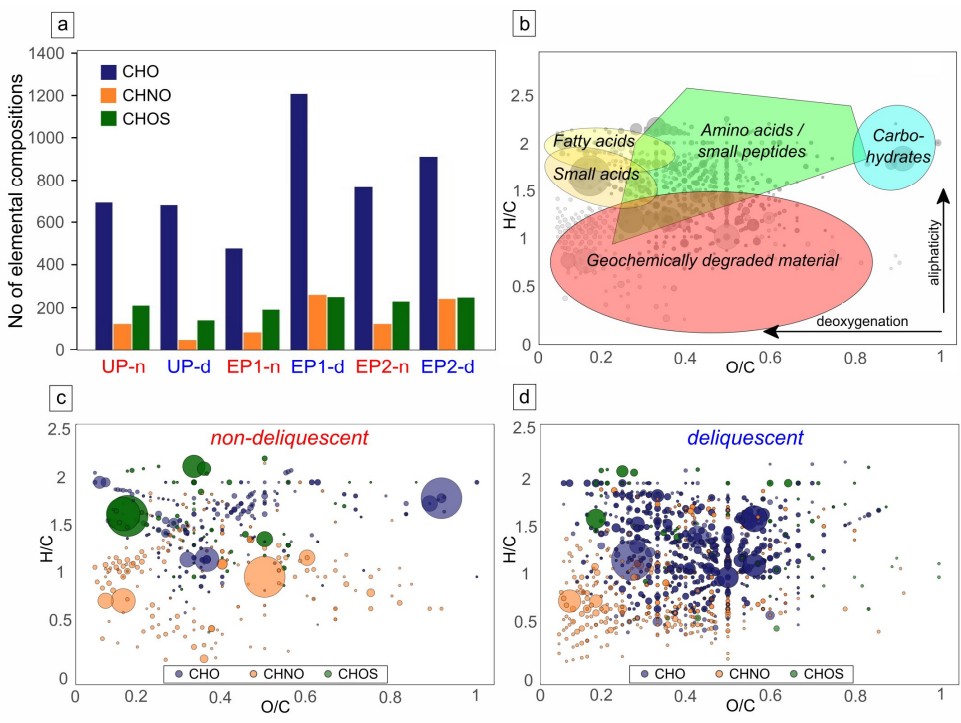

Figure 7: Compositional profiles of organic matter. a) Abundance of elemental compositions in uneroded and eroded polygon sites. b) Exemplary van Krevelen diagram plotting the hydrogen to carbon atomic ratio (H/C) as a function of the oxygen to carbon (O/C) atomic ratio of organic compounds. The positions of chemical classes (colored areas) are depicted in compositional space. Highly aliphatic compounds are mostly presented in the upper (H/C ratio > 1) and aromatic compounds in the lower area (H/C ratio < 1). c) Molecular compositions specific for non-deliquescent surfaces and d) for deliquescent, nitrate-rich surfaces are plotted as CHO (blue), CHOS (green), and CHNO (orange), and bubble sizes depict mass signal intensities.

The relationship between the atomic ratio O/C vs. H/C of the assigned molecules is plotted in the van Krevelen diagrams (Fig. 7b-d, S5). The results revealed a broad distribution within the compositional space reflecting the complexity of the organic molecules contained in the samples encompassing possible amino acids, small peptides, and phenolic compounds. The dominance of phenolic compounds reflects an overall geochemical signature, indicating low bioactivity and long-term geochemical processes responsible for lignin-like organic matter



degradation. Profiling the mass signal intensities across the entire spectrum reveal a differentiation of samples into
two groups: non-deliquescent soils show only minor specific molecules with few intense CHOS signals (Fig. 7c),
whereas deliquescent soils with additional chlorides and nitrates (especially from EP1 and EP2 sites) have more
specific CHO and CHNO molecules (Fig. 7d).

## 4 DISCUSSION

### 4.1 Deliquescence-driven environment

The investigated sites are located on alluvial fan deposits of Miocene to Pliocene age (Sernageomin and others,
2003; Amundson et al., 2012). During millions of years of hyperaridity large amounts of atmospherically derived
salts, including nitrates, were added by dry deposition (Ericksen, 1981; Michalski et al., 2004; Ewing et al., 2006).
Although erosion is generally minimal in the Atacama Desert, in few locations, vulnerable to eolian erosion, the
upper soil layers have been removed (Sager et al., 2022). This erosion was evident at the EP sites, indicated by the
highly soluble salts and the anhydrite at the surface of the polygons, both found otherwise in the subsurface below
40 cm depth of the uneroded soils (Schulze-Makuch et al., 2018; Arens et al., 2021; Sager et al., 2021). Local
morphology and topography did not indicate a connection to active fluviatile channels (Fig. S1). However, the
erosional surfaces tend to correlate with topographic lows, such as ancient channels and the valley basin (Fig. 1b).
These ancient morphological features have been shown to influence soil composition and structures subsequently
impacting the vulnerability of the soil surface to eolian erosion (Pfeiffer et al., 2021; Sager et al., 2022).
Due to this erosion, the exposed hygroscopic nitrate- and chloride-salts interact with occurring rain, fog, and even
increased air humidity. Generally, minimal precipitation occurs only once every few years (McKay et al., 2003;
Bozkurt et al., 2016). In contrast, air humidity fluctuates diurnally from values as low as 5 % RH during the day,
to high values reaching saturation during the night due to strong temperature fluctuations. This can also lead to fog
formation. Normally, the dew point on the surface is not reached solely by a drop in temperature (McKay et al.,
2003), but also due to the presence of hygroscopic salts that enable deliquescence, providing liquid water even at
RH >75 % for halite and >74 % of nitratine at 20 °C (Greenspan, 1977). For eutectic NaCl-NaNO$_3$ mixture
deliquescence occurs even at 67 % RH (Tang and Munkelwitz, 1994; Gupta et al., 2015).
The repeated cycles of moistening and evaporation of the hygroscopic soil patches can create efflorescence
structures, like soil doming and encrustation of salt-rich sediment (Sager et al., 2022), which are also observed at
the EP2 site. The absence of the efflorescence at the EP1 site correlates with the lower water uptake of the soil,
while the salt content is similar (Fig 3,4). This suggests salt exposure at EP1 may have occurred more recently and
that the secondary processes have not yet caused measurable effects. Additionally, the increased moisture uptake
of EP2-d compared to EP1-d suggests that the surface morphology has an impact on the deliquescence. Possibly,
due to the efflorescence structures (Fig. 2) the soil surface may cool down more efficiently, lowering the dew
point.
The ongoing process of deliquescence and efflorescence of the surface at EP2 could also be responsible for the
higher abundance of phyllosilicates and carbonates compared to EP1. These may have accumulated through the
entrapment of eolian dust, sticking to the moist soil surface and incorporated into the salt crust. Alternatively, the
phyllosilicates and carbonate can have formed autochthonously due to more frequent presence of water in these
soil patches resulting in enhanced aqueous weathering (Ewing et al., 2006).

### 4.2 Habitability of the salt crust

With the common notion "follow the water" in searching for life, the repeated occurrence of soil moisture was a
strong indicator of a new potential micro-habitat in the hyperarid Atacama Desert. The environmental monitoring
and the geochemical results confirmed the initial observation in the field that the soil surfaces can provide moisture,
which is potentially suitable for microbial activity (Stevenson et al., 2015). Deliquescence prolongs the presence
of liquid water, making microbial activity more likely. This is crucial, considering that moisture, mainly brought
into the Yungay valley by humid air from the Pacific Ocean, is only sufficient to yield ~400 h per year with dew
formation (>95 % RH) (Warren-Rhodes et al., 2006). Extrapolating the observed deliquescence during the
sampling campaign (with RH >85 %) and the recording of air humidity over two years, the duration of moist soil
is ~10 times longer compared to surfaces with no hygroscopic salts.



However, our microbiological analysis did not support an enhanced habitability for microorganisms of the investigated soils. In contrast, the results showed even lower microbial activity and microbial growth compared to the control samples with no observed deliquescence and no or minor amounts of hygroscopic salts (Fig. 3, 5a). For the cell cultivation experiments a low salinity growth medium was used, which could have favored the growth of microorganisms in the non-deliquescent soil samples or could have suppressed halophilic organisms. For future investigations, additional experiments with more saline growth media could help to verify this trend. The genetic data of the cultivated bacteria indicates that these are native organisms known from the Atacama Desert specifically in the Yungay valley (Navarro-Gonzalez et al., 2003; Azua-Bustos et al., 2019; Azua-Bustos et al., 2020). On the other hand, the plant-symbiotic genus *Rhizobium* found in the deliquescent soil samples is unlikely to thrive in the unvegetated study area (Araya et al., 2020). This and the lower bacterial abundance but higher alpha-diversity in the deliquescent samples may suggest that the deposition of airborne input of microorganisms is promoted by enhanced adhesion of moist soil surfaces.

Previous studies investigated non-deliquescent soils in the hyperarid region regarding their biological activity and diversity showing similar results to the here investigated non-deliquescent soils (Connon et al., 2007; Lester et al., 2007; Crits-Christoph et al., 2013; Schulze-Makuch et al., 2018; Warren-Rhodes et al., 2019; Knief et al., 2020; Shen, 2020; Sager et al., 2023). Also, metabolic signatures match, showing a geochemical footprint, superimposed by fresh organic material indicating at least some metabolic activity (Schulze-Makuch et al., 2018). Microhabitats previously studied and most related to the here investigated deliquescent soils are halite nodules within salars, which also undergo diurnal deliquescence (Wierzchos et al., 2006; Robinson et al., 2015; Valea, 2015; Schulze-Makuch et al., 2021; Perez-Fernandez et al., 2022). Spatially closest examples can be found in the Aguas Blancas Salar, 10 km east of the sample site. Besides microscopic confirmation of intact microorganisms, these niches show higher PLFA concentration and diversity (Ziolkowski et al., 2013; Schulze-Makuch et al., 2021), as well as metabolic composition reflecting fresh biological material and microbial activity (Schulze-Makuch et al., 2021).

Comparing the sulfate-rich shallow subsurface and halite nodules with our nitrate-rich salt crust the most striking difference is the nitrate abundance in the here investigated salt crusts. To our knowledge, only endolithic communities have been reported in salt crusts containing halite or gypsum (Wierzchos et al., 2006; Wierzchos et al., 2011).

The reduced habitability of the nitrate crusts can have multiple reasons. Potential organisms thriving in the formed brine saturated with $NaNO_3$ would be confronted with higher osmotic stress, due to high solubility of $NaNO_3$. Additionally, nitrate induces chaotropic stress affecting the bio-macromolecular structure (Lima Alves et al., 2015). This characteristic correlates in large parts with the Hofmeister series giving the order of effectiveness of protein precipitation as follows: $SO_4^{2-} < Cl^- < NO_3^- < ClO_4^-$ (Hyde et al., 2017). While microbial growth could not be detected yet in $NaNO_3$ solutions with concentrations exceeding 34 wt% (4.9 M) (Heinz et al., 2021), the brine formed by deliquescence would have an initial concentration of 10.9 M (i.e. saturation point at 25 °C) (Archer, 2000). Nitrates can also induce reactive oxygen species (ROS, e.g., $OH^-$, $H_2O_2$) or reactive nitrogen species (RNS, e.g., $NO^{\bullet}$, $NO_2^-$) which cause oxidative and nitrosative stress (Ansari et al., 2015). This can occur in the presence of UV radiation, which is intense in the high-altitude and cloud-free Atacama Desert reducing nitrate to nitrite and $OH^-$, or $NO^{\bullet}$ and $O_2^{2-}$ (Yang et al., 2021).

The nitrate-rich efflorescence crusts create an extremely rare environment. Sand wedge polygonal grounds are widely found in the Yungay valley and within the hyperarid core of the Atacama Desert (Ericksen, 1981; Sager et al., 2021). However, due to the hyperarid condition, erosion is minimal which is why these erosional surfaces are scarce. Despite the hyperaridity, the nitrate crust is presumably not stable at the surface, as precipitation is eventually washing the salts on the alluvial fan down into the subsurface or is eroded by the wind.Hence, the occurrence of nitrate-rich environments is likely so rare throughout Earth history that life has not evolved any strategies for adaptation to cope with these exceptionally harsh conditions.

## 4.3 Preservation of biomolecules

The here measured biological and biogeochemical parameters indicate that habitability is reduced in the nitrate-rich soil crusts. However, organic carbon is elevated in comparison to the surrounding soil as well as compared to previous studies (Connon et al., 2007; Lester et al., 2007). This is also indicated by the composition of organic matter, which was more diverse at the nitrate-rich sites. The uneroded caliche layer residing at depth, being the precursor of the deliquescent surfaces, does not show such an abundance and diversity of organic carbon (Fuentes et al., 2021; Schulze-Makuch et al., 2021). Thus, carbon compounds have been presumably introduced after



exposure to the atmosphere. As proposed for the phyllosilicates and carbonates (Sager et al., 2022), also organic
carbon could be trapped by the moist salt crusts in the form of airborne dust, including microbes or already
degraded organic matter. Potential sources for the organic matter could be the sea spray from the Pacific Ocean
transported by the dominating west wind (McKay et al., 2003; Azua-Bustos et al., 2019). Also, fog oasis and sparse
plant cover in the coastal range could be potential sources for more organic-rich dust particles (Quade et al., 2007).
Salts are recognized for their role in stabilizing biomarkers. Hypersaline environments often exhibit enhancements
of particular molecular biomarkers, such as gammacerane (Damsté et al., 1995), or a higher ratio of acidic to basic
amino acids (Rhodes et al., 2010) and lead to entrapment of biogenic molecules (Cockell et al., 2020) and microbes
(Perl and Baxter, 2020). Nitrate salts are known to inhibit microbial activity and have been used to cure food,
especially meat (Majou and Christieans, 2018). Besides higher dust (including organic matter through organic
aerosol dry deposition) accumulation rates, biological degradation could also be hindered in the same way by the
presence of nitrates, leading to higher TOC values in the here investigated nitrate-rich soil crusts. Nitrate-rich
subsurface layers within million-year-old hypersaline deposits of the Atacama Desert revealed a variety of
biomolecules, confirming the high biosignature preservation potential of nitrates (Fernández-Remolar et al., 2013).
Besides these benefits for biomass preservation, ROS or RNS originating from UV-exposed nitrates as discussed
earlier, can enhance geochemical degradation of biomolecules. Indications can be found in the profiles of organic
matter, where small CHNO species dominate across the nitrate crusts pointing to a geochemical breakdown of
organic molecules with reactive nitrogen species. However, in comparison to the surrounding non-deliquescent
soil surfaces, the nitrate-rich soils seem to promote the preservation of organic matter.

## 475  4.4 Indications for the search for life on Mars

In addition to abundant sulfate and chloride deposits also nitrates have been detected on Mars e.g., by Curiosity
Rover in the Gale Crater at concentrations up to 600 mg kg$^{-1}$ (Stern et al., 2015; Stern et al., 2017). Morphological
and geochemical indicators suggests that during the Hesperian and early Amazonian periods environmental
conditions like the water availability on Mars has been comparable to the contemporary Atacama Desert (Stepinski
and Stepinski, 2005; Bibring et al., 2006). It is plausible that like in the Atacama Desert, also on Mars the
accumulation of nitrates was dominated by dry fallout from the atmosphere, produced by volcanic lightning and
impacts during the first 1 Ga of Mars history (Michalski et al., 2004; Segura and Navarro-González, 2005;
Manning et al., 2009). Analogous to the Atacama Desert, nitrate deposits could have formed in the Martian
subsurface during that time. Extrapolating our findings to Mars would make nitrates-rich soils as a potential habitat
unfavorable, but due to the enhanced preservation of biomolecules these are still a promising target for finding
relics of ancient Martian life. This is also indicated by the detection of biomolecules in a million-year-old nitrate-
rich deposit in the Atacama Desert (Fernández-Remolar et al., 2013). The habitability of Martian nitrate-rich crust
should not be ruled out, since the evolutionary pressure on Mars could have enabled microbes to adapt to high
nitrate concentrations, as life on Earth has adapted thrive in brines containing the most abundant salt, being NaCl
(Heinz et al., 2019). Due to the gradual and global expansion of hyperarid conditions on Mars, putative life could
have evolved strategies to adapt to high salt concentrations, including nitrates, and by making use of their
hygroscopic nature (Davila and Schulze-Makuch, 2016; Maus et al., 2020). Maybe even more important on Mars,
these nitrate deposits could also represent a rare nitrogen-source for life as we know it, to build biomolecules like
amino acids and nucleobases.

## 495  5 CONCLUSION

Our investigation of the deliquescence of nitrate-rich soils in the Atacama Desert provides new insights into the
dynamics and the habitability in one of the Earth's most extreme environments. Despite providing transient
moisture, our results indicate that the nitrate-rich surfaces exhibit lower microbial abundance and activities
compared to the surrounding non-deliquescent surfaces. The high nitrate concentrations appear to suppress
microbial activity, likely due to osmotic and chaotropic stress and the potential production of reactive nitrogen
species. Remarkably, the nitrate-rich soil surfaces bear elevated geochemically degraded organic matter, indicating
an enhanced biomolecule preservation of these environments under such extreme conditions. These findings
highlight the dual role of nitrates in organic matter preservation and microbial inhibition. The inhabitability despite
water availability and the preservation potential in nitrate-rich soils underscores their importance in the search for
life in hyperarid environments on Earth and aids in the field of astrobiology to the search for life on Mars.



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

Database. Scientific Data 9, 409.

**Acknowledgements**

We thank Yunha Hwang for supporting us during the fieldwork. We would like to thank the following people for
their contributions to this work: Manuela Alt and Kirsten Weiß from the HU Berlin for conducting the elemental
analysis; Ferry Schiperski and Thomas Neumann for the access to their laboratories at the Institut für Angewandte
Geowissenschaften at the TU Berlin; Maria Scharfe and Eckhard Flöter from the Institut für
Lebensmitteltechnologie und Lebensmittelchemie at the TU Berlin for using their laboratory equipment. Landsat-
8 image courtesy of the U.S. Geological Survey. We acknowledge support by the European Research Council
Advanced Grant Habitability of Martian Environments (#339231).

**Competing interests**

The authors declare no competing interests.

**Data availability**

The authors declare that all the data supporting the findings of this study are available within the article and its
Supplementary Information file, or available from the corresponding author on request. Sequence data that support
the findings of this study will be deposited in the European Nucleotide Archive with the primary accession code
PRJEB70476.

**Author contributions**

F.A.: conceptualization, fieldwork, sample preparation, XRD measurement, water analysis, ATP analysis, data
evaluation and visualization, manuscript writing; A.A.: conceptualization, fieldwork, data evaluation, manuscript
writing; C.S.: fieldwork, PLFA measurements, manuscript writing; H.P.G.: genomic data evaluation; K.M.:
PLFA data evaluation; R.M.: ATP data evaluation; M.P.: ATP measurement and data evaluation; P.S.K.: organic
matter data evaluation; J.U.: organic matter measurement and data evaluation; B.V.: conducting cultivation
experiment and genomic analysis; P.Z.: cultivation experiment data evaluation; L.Z.: genomic analysis and data
analysis; D.S.M.: project supervision; all authors modified and revised the manuscript.