# Peer review of "Microbial response to deliquescence of nitrate-rich soils in the"

_EGUsphere, 2024_

## Author Response (AR1)

Dear reviewers,

Thank you very much for your valuable and constructive review. Below we provide a point-by-point answer in green to the concerns and questions raised.

*Reviewers comments*

**RC1**: 'Comment on egusphere-2024-1859', Anonymous Referee #1, 21 Aug 2024  reply

Thank you for the thoughtful and constructive review. We appreciate your positive feedback.

This study focusses on deliquescent soil surfaces in the Atacama Desert to evaluate their potential as habitat for microorganisms. While using a range of different proxies they gained important information that hygroscopic salts provide lower microbial abundances and that their activities are lower than in non-deliquescent soil surfaces which leads them to the conclusion that high nitrate concentrations suppress the microbial activity but preserve eolian-derived biomolecules.

Although in the last decades the Atacama was already intensively studied regarding its habitability, nitrate rich soil crusts have been neglected so far. Therefore, this study provides an important value for the current state of the art regarding the habitability of the Atacama Desert and also other extraterrestrial surfaces e.g., on Mars.

I recommend the publication of this study in Biogeosciences following some minor revisions.

In the Introduction, I was missing a passage about the night and day shift which become very important when reading the rest of the paper e.g., the results part of it.

We added a passage regarding day-night shifts.

L29: Could you add a sentence to better point out the current research gap (e.g., habitability of salts in the Atacama Desert or what is the significance of nitrates for microbial life in one of the most arid regions on Earth, serving as an outstanding Martian analog?)

We added a sentence on the deliquescence habitats (halite crusts) in the Atacama and the research gaps on other salt crusts (e.g., nitrate-rich crusts)

L29: Are the words "newly discovered" so important to mention at this point? A potential reader maybe does not know directly that there are already many investigations before.  If there are so many could you point them out a bit more clearly in the Introduction part?

With the additional sentence, "newly discovered crusts" makes more sense.

L46: can you replace "absolute desert" with "hyperarid core" which you mention later to keep it more consistent?

Of course, we changed it to hyperarid core.

L57: the one of the last islands….is found or one of the last islands are found. Next sentence also starts with "these" so refers to a plural form.

Corrected to "The last islands of habitability".

L59: of salts

corrected

L64: write salars also in italic

Changed accordingly.

Figure 1: a) replace "Sanitago" with "Santiago"

Thanks! Changed accordingly.

L86: can lead to the exposure of the salt-rich subsurface or can lead to the exposure of salt-rich subsurfaces

Changed to "subsurfaces"

87: Similar to halite nodules, salt-encrusted surfaces can form. These are composed of sulfate, chloride and nitrate salts and develop a similar efflorescence morphology.

Changed accordingly

Section 2.1

Am I right that you had 3 sampling sites, where you sampled the a) deliquescence affected area and b) non deliquescence affected area. Then in these 3x2 sites you took 3 samples in the morning for geochemical analysis and 3 for biological samples and the same in the evening?! Did you also disinfect your shovel or sampling equipment?

Correct, we disinfected your equipment before and between every sampling and started with the biological samples to minimize contamination.

Would it not also have been nice to repeat that sampling for at least 2 or 3 days for having a better data base.

Definitely, however the sampling was conducted in a framework of a wider field campaign with a tied schedule which did not allow more sampling of these sites.

As Temperature and RH of the air was measured in 2018 and 2019, I guess the samples were also taken during that time? Can you add the time as well, please.

We added the date and time in this section. You can find them also in figure 4a.

What about statistics? Can you add something about it please?

Temperature and RH average over the course of the measurement were added.

L137: which standards did you use? And what about blanks?

Glutamic acid was used as reference standard. For both methods 14 blanks each were measured to determine the limit of quantification.

L161: was conducted

"PLFA extraction and subsequent analysis were conducted…" should be correct.

For the PLFAs did you also measure a standard for comparison and identification?

PLFAs were identified according to their chromatographic behavior compared to a mixed fatty acid standard (containing the usual saturated, unsaturated and branched fatty acids) and/or their characteristic mass spectra. To quantify the PLFAs, we added a deuterated phospholipid standard (PC$_{54}$, phosphatidyl choline with two deuterated tetradecanoic ester side chains) as internal standard after the lipid extraction.

L161: was conducted

Plural form should be correct.

You said before that you monitored RH and and temperature between 2018 and 2019 but only show the data during the time of sampling which is great, but I was wondering how to categorize it during this long time series. Where are we here just in a good mean of the time?!

We added more information regarding the temperature and RH.

L270: Why did you only analyze the samples taken during the morning for the geochemical analyses?

In contrast to ATP, being an indicator for metabolic activity which potentially might change throughout the day, phospholipids are an integral part of the cell membranes of living microorganisms. Short-term changes throughout the day are not expected, since membranes are usually well-adapted to the prevailing environmental conditions. Therefore, we only measured the PLFA data in the samples gathered in the morning.

L395: When you say "follow the water" for the search of life and there is a difference between day and night why did you measure e.g., PLFAs only in the morning?

See the answer before. The short-lived ATP did not show any changes during the day-night cycle.

L446: wind.Hence

corrected

**RC2**: '_Comment on egusphere-2024-1859_', Anonymous Referee #2, 09 Sep 2024

Thank you for your valuable and concise feedback. We appreciate your positive remarks and suggestions.

Line 37: specify how you know the organic matter has been degraded

Indicated by the molecular composition.

Line 61-63: Excellent comparison. Consider moving earlier in the text.

Changed accordingly.

Line 95,96: while it may provide better preservation than the surface, there are still irradiation considerations. Note the harmful issues that biology would still need to overcome on the surface

Correct, we added the caveat and that these deposits are good candidates to search for evidence of life close to the surface, which are most accessible.

Line 117, 255: You mention Aw but don't state any values (<0.5) until line 255. Were these the only values recorded?

Line 117 is the method section. The Aw results are presented in the result section. All Aw measurements are presented in figure 4.

Fig 6 and Lines 331 and onwards: These were sequenced at the genus level. Given the mineralogy in Fig 5 these don't need to be "fully" halophilic given that halite and gypsum are in small amounts in the samples. Can you comment more on the halophilic characteristics with respect to preservation and the lack of salts + how that would effect the deliquescence?

The sequence results compare only the EP2 sites, where the deliquescence site has substantial amounts of hygroscopic salts (~10 wt % Cl- + NO3-). When deliquescence occurs, the brine (the potential microhabitat) is saturated resulting in even higher salt concentration.